# Modelling and Forecasting Temporal PM$_{2.5}$ Concentration Using Ensemble Machine Learning Methods

**Obuks Augustine Ejohwomu** [1,*] , **Olakekan Shamsideen Oshodi** [2], **Majeed Oladokun** [3],
**Oyegoke Teslim Bukoye** [4], **Nwabueze Emekwuru** [5], **Adegboyega Sotunbo** [6] **and Olumide Adenuga** [6]

[1]  Department of Mechanical, Aerospace & Civil Engineering, The University of Manchester, Oxford Rd, Manchester M13 9PL, UK
[2]  School of Engineering and the Built Environment, Anglia Ruskin University, Chelmsford CM1 1SQ, UK; ooshodi1@gmail.com
[3]  School of Science, Engineering and Environment, University of Salford, Salford M5 4NT, UK; lyday011@gmail.com
[4]  School of Management, University of Bath, Claverton Down, Bath BA2 7AY, UK; otb34@bath.ac.uk
[5]  Institute for Future Transport and Cities, Coventry University, Priory St, Coventry CV1 5FB, UK; ab9992@coventry.ac.uk
[6]  Department of Building, University of Lagos, Lagos 101017, Nigeria; gboyega90@yahoo.com (A.S.); oaadenuga@yahoo.com (O.A.)
*  Correspondence: obuks.ejohwomu@manchester.ac.uk

**Abstract:** Exposure of humans to high concentrations of PM$_{2.5}$ has adverse effects on their health. Researchers estimate that exposure to particulate matter from fossil fuel emissions accounted for 18% of deaths in 2018—a challenge policymakers argue is being exacerbated by the increase in the number of extreme weather events and rapid urbanization as they tinker with strategies for reducing air pollutants. Drawing on a number of ensemble machine learning methods that have emerged as a result of advancements in data science, this study examines the effectiveness of using ensemble models for forecasting the concentrations of air pollutants, using PM$_{2.5}$ as a representative case. A comprehensive evaluation of the ensemble methods was carried out by comparing their predictive performance with that of other standalone algorithms. The findings suggest that hybrid models provide useful tools for PM$_{2.5}$ concentration forecasting. The developed models show that machine learning models are efficient in predicting air particulate concentrations, and can be used for air pollution forecasting. This study also provides insights into how climatic factors influence the concentrations of pollutants found in the air.

**Keywords:** ensemble machine learning methods; modelling and forecasting; PM$_{2.5}$; predictive performance

## 1. Introduction

In the developing world, the number of people living within urban areas is rapidly increasing. For instance, it is projected that 68% of the global population will reside in urban areas by 2050 [1]. Based on census data and projections, the population of Lagos was 9.1 million in 2006, and it was projected that the city would have 24.3 million residents in 2015 [2]. This growth creates lots of social, economic, and environmental problems. Urban heat islands [3] and the rise in air pollution [4]) are some of the consequences of this rapid urbanisation. According to previous studies [5,6], the increase in the number of extreme weather events and healthcare costs, among others, have been linked to the air pollutants emitted as a result of the activities of city dwellers. To protect the planet, there is an urgent need for stakeholders (such as policymakers) to develop and implement strategies for reducing these air pollutants.

The outcomes of research focused on monitoring, modelling, and predicting air pollutants provide evidence for policy interventions, and justify changes in human behaviour.

Traditionally, regression-based methods have been used for forecasting air pollutant concentration. For instance, the logistic regression model was used for forecasting the concentration of $NO_2$ in a study by Drye [7]. However, the need to improve the accuracy of forecasts has led to the application of other methods in recent studies. Neural networks [8], Prophet [9], and support-vector machines [10] are examples of methods used in recent studies on air pollutant forecasting. These methods have inherent weaknesses that affect the quality of their forecasts. For example, regression-based methods cannot adequately capture nonlinearity present in real-world data [11]. McKendry [12] used linear and nonlinear methods for air pollutant forecasting, and found that the nonlinear models (i.e., neural networks) produce a better forecast. Moreover, other studies have shown that neural network models are susceptible to overfitting and local minima [13]. Due to the identified limitations, there is a need to evaluate the effectiveness of newer modelling techniques that have emerged as a result of advancements in data science.

Prophet and ensemble models are examples of new algorithms used for solving problems relating to prediction. Research has shown that the ensemble model, which can combine a variety of models, tends to generate more accurate forecasts when compared with standalone models [14,15]. The present study seeks to examine the effectiveness of using ensemble models for forecasting the concentrations of air pollutants, using $PM_{2.5}$ as a representative case. A comprehensive evaluation of the ensemble methods was carried out by comparing their predictive performance with that of other standalone algorithms. This goal was achieved by addressing two objectives: (1) to model and forecast $PM_{2.5}$ using seven algorithms (autoregressive integrated moving average, exponential smoothing, Prophet, neural networks, random forests, support-vector machines, and extreme gradient boosting), and (2) to apply the ensemble models for the forecasting of $PM_{2.5}$ concentrations.

The identification of reliable models for forecasting of $PM_{2.5}$ concentrations is important for several reasons. First, the information from such models can be used to reduce the exposure of humans to air pollution (e.g., people can use face masks on days when the expected exposure is high). Second, the models can be used to investigate the effects of policy on the reduction in the level of air pollution (e.g., what is the impact of low-emission zones on air pollution in cities?). Finally, the models provide insights into the variables that have significant effects on the concentration of $PM_{2.5}$.

*Significance of $PM_{2.5}$*

Modelling and forecasting of air pollutant concentrations have been the focus of previous research. The pollutants that were modelled in those studies include $PM_{2.5}$ [9], $PM_{10}$ [16], and $NO_2$ [7], among others. $PM_{2.5}$ has received more attention due to the link between its concentration and various health problems, such as heart conditions, respiratory illnesses, and neurological disorders, among others [17,18]. Other studies have established a positive relationship between exposure to $PM_{2.5}$ and the need for hospital visits [18,19]. Taken together, the concentration level of $PM_{2.5}$ has an adverse effect on human health and the costs of providing healthcare. Thus, the development of reliable tools for forecasting of $PM_{2.5}$ is vital for establishing effective strategies for reducing exposure to it.

## 2. Methods

### 2.1. Study Area and Data

Lagos was the former administrative capital of Nigeria, and it remains the hub of economic activities within the country. The population of the city keeps increasing, and it is projected that it is approaching 'megacity' status [20]. High population density, economic activities, and transportation generate large volumes of air pollutants, such as $PM_{2.5}$ [21]. For instance, due to the shortfall in power supply, business owners spend a huge chunk of their operating costs on the running of fossil-fuel-powered generators that generate noise and air pollution [22]. Moreover, Zeng et al. [23] showed that the concentration of $PM_{2.5}$ is Lagos is higher than those recorded in Hong Kong. In the study presented in this

paper, PM$_{2.5}$ data collected over 15-min intervals over a 6-month period at a location at the University of Lagos were used to assess the effectiveness of using contemporary techniques for modelling and forecasting the concentration of PM$_{2.5}$.

The concentration of PM$_{2.5}$ was measured using the EarthSense Zephyr air quality sensor; meteorological variables (temperature and relative humidity) were also quantified. These data (PM$_{2.5}$ concentration and the meteorological variables) were used to train the univariate and multivariate models developed in this study. Subsequently, the trained models were used to generate forecasts of PM$_{2.5}$ concentration.

## 2.2. Modeling Techniques

As stated previously, seven methods were used to predict the PM$_{2.5}$ concentration. A regression-based method—i.e., autoregressive integrated moving average (ARIMA)—was applied in this study because it has been extensively applied in previous research on air pollution forecasting [24]. However, the ARIMA model does not have the capacity to capture nonlinear relationships. Therefore, the predictive performance of machine learning methods (such as support-vector machines, neural networks, and random forests, among others) was compared with the ARIMA model. The nonlinear relationships between the variables included in machine learning models are difficult to interpret. However, these models can be interpreted through the use of sensitivity analysis (i.e., the improvement in predictive accuracy through the inclusion or removal of certain variables).

ARIMA: ARIMA is a time-series modelling approach used for forecasting. The ARIMA model has three elements, i.e., an autoregression model, a moving average model, and differencing [25]. ARIMA models strive to capture information present in the autocorrelation of the data and use it for modelling and forecasting purposes. The ARIMA model can be mathematically expressed as:

$$y'_t = c + y'_{t-1}\varnothing_1 + \ldots + \varnothing_p y'_{t-p} + \theta_1 \varepsilon_{t-1} + \ldots + \theta_q \varepsilon_{t-q} + \varepsilon_t \tag{1}$$

where $y'_t$ represents the differenced time-series data for PM$_{2.5}$ concentration, $c$ is the constant, and $\varepsilon$ is the error term. The order of differencing of the time-series data is 'd'. The 'predictors' on the right-hand side of Equation (1) include the lagged values of PM$_{2.5}$ concentration and lagged errors. The $p$, $d$, and $q$ elements of the ARIMA model are automatically determined using a variant of the Hyndman–Khandakar algorithm [26].

Exponential smoothing: This method was introduced in the late 1950s [27–29], and has been applied to forecasting problems in several fields; it is an extension of the simple moving average system in which past observations are weighted equally based on the averages of a number of subsets of the full dataset; for the exponential smoothing method, the weighting assigned to past events exponentially decreases over time. The formula for the exponential smoothing can be expressed as:

$$y_{t+1} = F_t + \alpha(y_t - F_t) \tag{2}$$

where $y_{t+1}$ is the forecast of PM$_{2.5}$ concentration at the next time period, $F_t$ is the forecast of PM$_{2.5}$ concentration at time ($t$), $y_t$ is the actual value of PM$_{2.5}$ concentration at time ($t$), and $\alpha$ is a weight called the exponential smoothing constant ($0 \leq \alpha \leq 1$).

Exponential smoothing has been used for modelling and forecasting of several problems, e.g., emergency department visits [30] and electricity consumption [31], among others. However, the weights attached to these previous values decay exponentially with respect to time [26]. Thus, the weights attached to recent observations (such as $y_{t-1}$) are higher than those of the older observations (such as $y_{t-6}$).

Prophet model: This is one of the modelling techniques applied in the current study. According to Hyndman and Athanasopoulos [25], the Prophet model is a nonlinear regression model that was introduced by Facebook [32]. The Prophet model can be expressed in mathematical form as shown in Equation (3). This method works best with data that

have 'strong seasonality'. For example, this technique has been used for the forecasting of groundwater levels [33] and variations in oil production [34].

$$y_t = g(t) + s(t) + h(t) + \varepsilon_t \tag{3}$$

where $g(t)$ describes a piecewise linear trend, $s(t)$ describes the various seasonal patterns, $h(t)$ captures the holiday effects, and $\varepsilon_t$ is a white noise error term.

Neural network (NN): Neural networks are a modelling technique that is inspired by nature. The neural network model mimics the human brain, and captures complex nonlinear relationships between dependent and independent variables. Two types of neural network models were used in this study: (1) neural network autoregression (NNAR), and (2) a neural network model based on multiple variables. The NNAR model is similar to the ARIMA model in terms of the variables included in the model. The NNAR model is a neural network model with 3 layers, i.e., 1 input, 1 hidden, and 1 output. For instance, an NNAR (9, 4) model is a neural network model that predicts $y_t$ using $(y_{t-1}, y_{t-2}, \ldots, y_{t-9})$ as inputs and 4 nodes in the hidden layer. The number of lags and number of nodes to be included are determined automatically as described in [25].

Several variables, including meteorological factors (see Table 1) and the $PM_{2.5}$ concentration, are included in the second neural network model. The second type of neural network model uses the information contained in previous values of the $PM_{2.5}$ concentration and independent variables to predict future values of the $PM_{2.5}$ concentration.

**Table 1.** Summary statistics of collected data from 1 December 2020 to 28 May 2021.

| Variable | Unit | Before Data Cleaning | | | After Data Cleaning | | |
|---|---|---|---|---|---|---|---|
| | | Range | Mean | SD | Range | Mean | SD |
| $PM_{2.5}$ | $\mu g/m^3$ | (1.73, 165.3) | 22.4 | 11.9 | (3.39, 65.96) | 21.9 | 9.8 |
| RH | | (13.1, 90.7) | 66.7 | 14.3 | (14.65, 89.4) | 66.7 | 14.2 |
| Temp | °C | (23.0, 43.0) | 32.2 | 4.2 | (24, 42.4) | 32.2 | 4.1 |

Note: SD = standard deviation; RH = relative humidity; Temp = temperature.

Random forest (RF): Random forest is an ensemble learning method that generates a large number of decision trees during the training process. The bootstrapping of the training dataset ensures that each decision tree that forms part of the random forest is unique. The prediction generated by the random forest model is an average of the output of each of the decision trees (see Equation (4)). Random forests prevent overfitting of the model to the training data. Reduced likelihood of overfitting is achieved through the integration of trees of various sizes and the averaging of the results.

$$\rho = \frac{1}{N} \sum_{n=1}^{n} N \tag{4}$$

where $\rho$ is the random forest prediction, and $N$ is the number of runs over the trees in the random forest.

The forecast generated by the random forest model is the average of the outputs of each decision tree.

Support-vector machine (SVM): The SVM was initially developed by Vladimir Vapnik and his colleagues [35]. The main goal of the SVM algorithm is to identify an optimal hyperplane (the formula for the hyperplane is presented in Equation (5)) that linearly separates the collected data into two groups.

$$w^T x + b = 0 \tag{5}$$

where $w$ is the weight vector, $x$ is the input vector, and $b$ is the bias.

The SVM algorithm was initially used for solving classification problems; however, it was extended with the adoption of the $\varepsilon$-insensitive loss function, and it can be applied to regression tasks (i.e., prediction of numbers) [36]. The SVM model uses nonlinear mapping to project input vectors in a higher dimensional space. The SVM model is able to always converge to optimal weights, and this gives SVM an advantage over the neural network model. In this study, the radial basis function (RBF) kernel was utilised, and the tuned hyperparameters of the SVM model are $\gamma$ and C.

Extreme gradient boosting (XGBoost): XGBoost is a relatively new machine learning algorithm that is an ensemble of decision trees that utilise the gradient boosting framework. The method was developed by Chen and Guestrin in 2016 [37]. The main intuition behind the XGBoost model is that the ensemble combines the output of a large number of decision trees to produce better predictions. XGBoost is an ensemble of gradient boosted decision trees. Moreover, the gradient boosting technique, which is a method introduced by Friedman [38], builds an ensemble of models through an interactive process aimed at improving accuracy—it initializes with a single model; subsequently, new models, which learn from the errors of the previous model, are added to the ensemble to improve the accuracy of the forecast. The predictions from the XGBoost model are computed using an additive strategy:

$$\hat{y} = \sum_{n=1}^{n} f_n(x_i) \tag{6}$$

where $x_i$ is the test sample, $i$ is the number of samples, $f_n$ is the $n$th tree model, and $n$ is the number of all trees in the model.

*2.3. Modeling Process*

The proposed models were used to generate predictions of the average hourly concentrations of $PM_{2.5}$. The data used to estimate the models were collected using the air pollution sensor described in the previous section. To evaluate the relevance of the metrological variables, two groups of models (i.e., with and without metrological variables) were developed. The ARIMA model was used as the baseline model. The decision to apply the ARIMA model was informed by its extensive application in previous studies on air pollution forecasting [39]. The forecasting performances of the other modelling techniques were compared with those of the ARIMA model. The modelling process was carried out in 4 stages: (1) data cleaning and preparation, (2) fitting the data to the models, (3) using the estimated model for forecasting, and (4) forecast evaluation. All of the models were implemented using the R programming software [40] and several packages, including *modeltime*, *modeltime.ensemble*, *timetk*, *tidyverse* [41,42], and *tidymodels* [43], among others. This software and these packages provide a platform for the application of statistical and machine learning models.

Data cleaning and preparation: The air pollution measuring device collected the $PM_{2.5}$ concentration and meteorological data at 15-min intervals. These data were aggregated and converted into hourly time-series data on $PM_{2.5}$ concentration, temperature, and relative humidity. As shown in previous research, the data cleaning process can have an impact on the accuracy of forecasting [44]; thus, the linear interpolation approach was used to replace the outliers that existed in the collected data. The time-series data on $PM_{2.5}$ concentration are presented in Figure 1. The descriptive statistics for the $PM_{2.5}$ concentration and metrological data are presented in Table 1.

The size of the dataset has a significant impact on the reliability of forecasts generated from models. To improve the accuracy of forecasting, the use of feature extraction methods (such as principal component analysis), dummy variables, Fourier series, and featuring engineering has been suggested in previous research [25,45,46]. Similarly, the outcome of forecasting competitions has shown that the use of feature engineering (e.g., extracting additional features out of the time data) improves the accuracy of forecasts. Therefore, in the present study, the '*modeltime*' package in the R programming software was used to create calendar-related features that were added to the multivariate models (i.e., models that

contain predictor and outcome variables). The variables incorporated into the multivariate models, along with their justifications, are described and explained in the Results section of this paper.

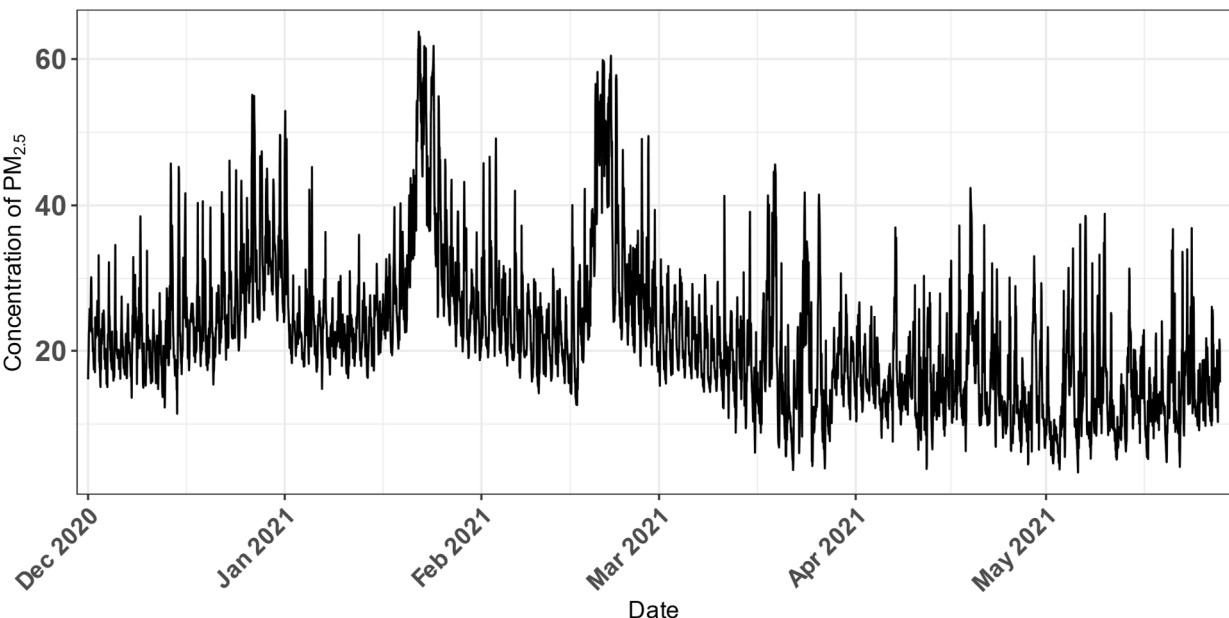

**Figure 1.** Time-series plot of PM$_{2.5}$ concentration data.

### 2.4. Forecast Accuracy

PM$_{2.5}$ concentration, relative humidity, and ambient temperature data covering the period between 1 December 2020 and 28 May 2021 (4283 data points) were collected and utilised for this study. The data were divided into two groups: training (4259 data points) and test (24 data points). The models were estimated using the training dataset. Subsequently, the trained models were used to generate forecasts for the previous 24 h, i.e., model validation. Adequate care was taken to ensure that there was no data leakage. To evaluate the accuracy of the forecasts generated by the models, the actual test data were compared with the forecast data.

Several metrics are usually used for evaluating the quality of the predictive accuracy of forecast models; they are usually classified into scale-dependent, percentage-based, relative-error-based, and relative measures. Hyndman and Koehler [47] identified the limitations of these four groups of metrics. For instance, it was stated that the mean absolute percentage error (MAPE) treats positive and negative errors in a different way [47]. Based on these limitations, the mean absolute scaled error (MASE) was identified as a suitable method for the evaluation of forecast accuracy. In the present study, MAE (mean absolute error), MASE, and root-mean-square Error (RMSE) were adopted. The use of RMSE is attributed to the extent of its usage in similar previous studies [48,49]. RMSE can be calculated as:

$$e_t = y_t - f_t \tag{7}$$

where $e_t$ is the forecast error at time $t$, $y_t$ represents the actual value of PM$_{2.5}$ concentration at time $t$, and $f_t$ denotes the forecast of $y_t$.

$$\text{RMSE} = \sqrt{\frac{1}{n}\sum_{i=1}^{n}\left(e_i^2\right)} \tag{8}$$

where $n$ is the total number of values.

MASE can be computed as follows [47]:

$$\text{MASE} = \frac{MAE}{MAE_{naive}} \tag{9}$$

where:

$$\text{MAE}(meanabsoluteerror) = \frac{1}{n}\sum_{i=1}^{n}|e_i|$$

and $MAE_{naive}$ is the MAR for the naïve model.

In forecasting research, it is expected that complex/advanced modelling techniques will generate better forecasts when compared with simple models, such as the naïve model. If a complex model produces a poor forecast when compared to a simple model, then it should be discarded [25]. If the value of MASE is greater than 1, this means that the model produces a forecast that is worse than that of the naïve model. A lower value (i.e., a value close to 0) of MASE indicates that the predictive accuracy of the model is better when compared with the naïve model.

## 3. Results

### 3.1. Exploratory Data Analysis

Correlation and autocorrelation are appropriate tools for measuring the relationships between variables and lagged values of time-series data, respectively. Correlation analysis was used to examine the relationship between the $PM_{2.5}$ concentration and the meteorological variables. As suggested by Hyndman and Athanasopoulos [25], Pearson's correlation was used for the bivariate analysis. The correlation plots and coefficients are presented in Figure 2. It was found that a weak negative correlation exists between $PM_{2.5}$ concentration, relative humidity, and temperature, with corresponding correlation coefficients of $-0.107$ and $-0.048$, respectively. In contrast, a strong negative correlation ($-0.929$) exists between relative humidity and temperature. Overall, all variables were negatively correlated with one another.

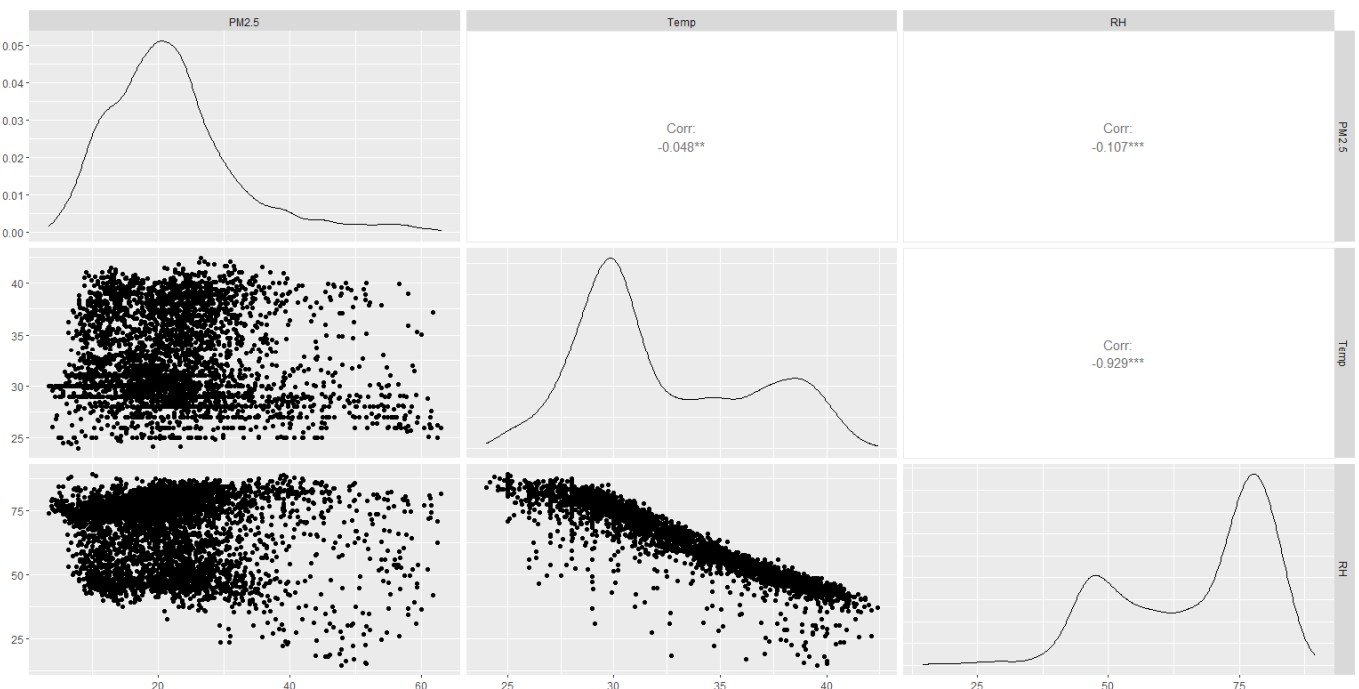

**Figure 2.** Scatterplot matrix of $PM_{2.5}$ concentration, temperature, and relative humidity.

The autocorrelation plot is presented in Figure 3. As can be seen from Figure 3, the values of the autocorrelations for lags 24, 48, and 72 are higher than those for other lags.

This observation can be attributed to the seasonal patterns present in the data. The peaks tend to be spaced 24 h apart. Various models were trained with these lag combinations to identify the appropriate one to be included in the multivariate models (for more details, see Section 3.3).

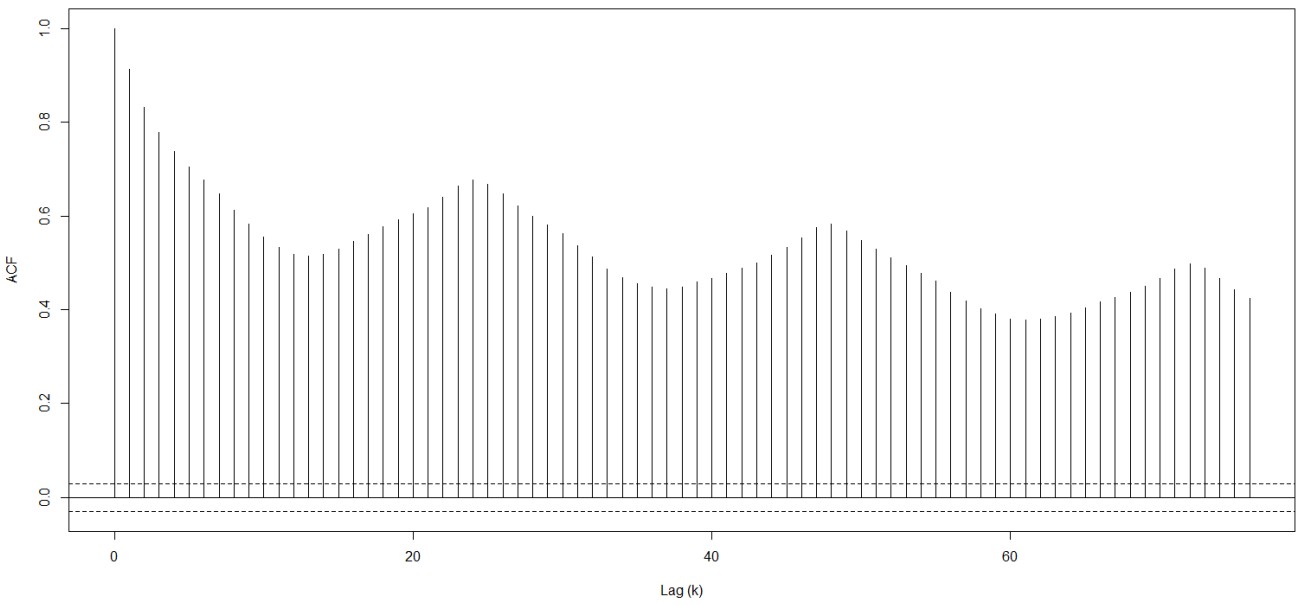

**Figure 3.** Autocorrelation plot of PM$_{2.5}$ concentration.

### 3.2. Univariate Models

Four univariate modelling techniques (i.e., ARIMA, exponential smoothing, Prophet, and NNAR) were estimated. The trained models were used to generate 24-hour forecasts of the PM$_{2.5}$ concentration values. The forecasts from the univariate models are compared with the actual concentration values of PM$_{2.5}$ in Figure 4. Additionally, the predictive performance metrics of the univariate models are summarised and presented in Table 2. The MAE, MASE, and RMSE of the ARIMA model were 1.83, 0.83, and 2.3518, respectively. Except for the NNAR model, all other univariate models outperformed the naïve model (i.e., the MASE values were lower than 1). From Table 2, it is evident that the values of MAE, MASE, and RMSE for the ARIMA model are lower when compared with those of the other univariate models. This finding shows that the ARIMA model outperforms the Prophet, exponential smoothing, and NNAR models when used for the forecasting of PM$_{2.5}$ concentration levels.

### 3.3. Multivariate Models

Multivariate models provide insights into the relationships that exist between independent and dependent variables. Five techniques (Prophet, XGBoost, SVM, RF, and neural network) were used to estimate various multivariate models. The factors considered when selecting variables to be included in the multivariate models included (1) literature, (2) availability of data, and (3) results from correlation analysis. The literature indicates that meteorological variables, such as humidity, are useful for the prediction of future concentration values of PM$_{2.5}$ [16,45]. Based on the availability of data, correlation analysis, and the literature, two meteorological variables (i.e., temperature and relative humidity) were added to the multivariate models as input variables.

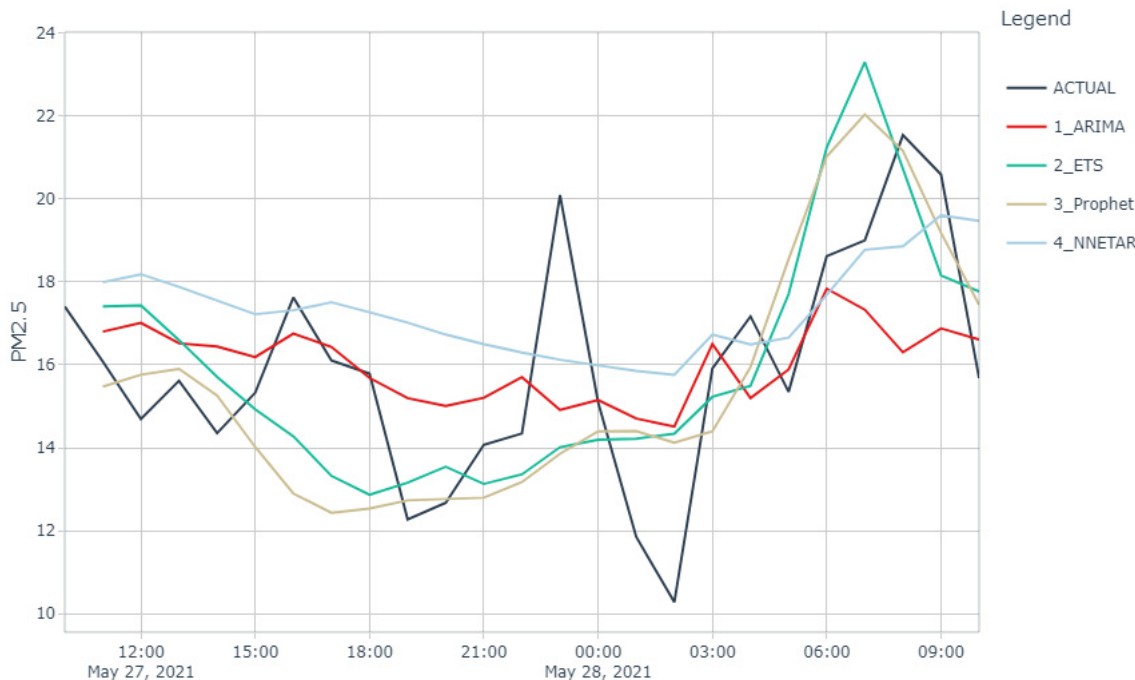

**Figure 4.** Forecasts from univariate models compared to actual values of PM$_{2.5}$ concentration; AC-TUAL: actual value of PM$_{2.5}$ concentration in the test period; ARIMA: predicted value of PM$_{2.5}$ concentration from the ARIMA model; ETS: predicted value of PM$_{2.5}$ concentration from the exponential smoothing model; Prophet: predicted value of PM$_{2.5}$ concentration from the Prophet model; NNETAR: predicted value of PM$_{2.5}$ concentration from the NNAR model.

**Table 2.** Predictive performance of the univariate models.

| S/N | Model | MAE | MASE | RMSE |
|-----|-------|-----|------|------|
| 1 | ARIMA | 1.82 | 0.83 | 2.3518 |
| 2 | Prophet | 1.96 | 0.89 | 2.4895 |
| 3 | Exponential smoothing | 2.08 | 0.95 | 2.4839 |
| 4 | NNAR | 2.29 | 1.04 | 2.7129 |

The autocorrelation plot (see Figure 3) shows that a seasonal pattern exists in the PM$_{2.5}$ concentration data. To capture this trend, four variables were identified: the 24th lag of PM$_{2.5}$ concentration, and 24 h, 48 h, and 72 h moving averages of PM$_{2.5}$ concentration. The '*modeltime*' package in the R programming software was used to create additional calendar-related features (see Table 3). The calendar-related features were based on the times at which the air quality data were collected (e.g., the day of the month). Research shows that calendar-related features are useful for the development of forecast models [40]. These calendar-related features capture the hourly, daily, weekly, and monthly patterns present in the PM$_{2.5}$ concentration data. The independent variables added to the multivariate models were (1) the meteorological variables, (2) lag (lags of the PM$_{2.5}$ concentration and moving averages) variables, and (3) calendar-related variables.

To identify the variables that have significant effects on the PM$_{2.5}$ concentration, various combinations of input variables were added to the multivariate models (see Table 4). All in all, 25 multivariate models were estimated and used for the forecasting of the PM$_{2.5}$ concentration values.

**Table 3.** Calendar-related features created using '*modeltime*'.

| Name | Type | Description |
|---|---|---|
| Index.num | Numeric | Time is converted into seconds (Base = 1970-01-01 00:00:00) |
| Month | Categorical (01–12) | Month of each air quality measurement (e.g., December = 12) |
| Month.lbl | Dummy | Month of the year for each air quality measurement |
| Day | Categorical | Day of each air quality measurement (13 December 2020 = 13) |
| Hour | Categorical (0–23) | Hour of each air quality measurement |
| Hour12 | Categorical (0–11) | Hour of the day on a 12 h scale |
| am.pm | Categorical (1–2) | Morning = 1 and Afternoon = 2 |
| Wday | Categorical (1–7) | Day of the week (Sunday = 1, Monday = 2, . . . , Saturday = 7) |
| Wday.lbl | Dummy | Day of the week |
| Qday | Categorical | Day of the quarter |
| Yday | Categorical (1–365) | Day of the year |
| Mweek | Categorical | Week of the month |
| Week | Categorical | Week number of the year |
| Week2 | Categorical | The modulus for biweekly frequency |
| Week3 | Categorical | The modulus for triweekly frequency. |
| Week4 | Categorical | The modulus for quadweekly frequency |
| Mday7 | Categorical (1, 2, . . . , 5) | The integer division of day of the month by seven (e.g., the first Saturday of the month has mday7 = 1 |

Source: Dancho, 2017.

**Table 4.** Variables included in the multivariate models.

| S/N | Input Variables | Description of Models |
|---|---|---|
| 1 | Calendar-related features | Prophet_time, XGBoost_time, SVM_time, NN_time, RF_time |
| 2 | 24th Lag of $PM_{2.5}$, and 24 h, 48 h, and 72 h moving averages of $PM_{2.5}$ + calendar-related features | Prophet_lag, XGBoost_lag, SVM_lag, NN_lag, RF_lag |
| 3 | Relative humidity + calendar-related features + lag features | Prophet_RH, XGBoost_RH, SVM_RH, NN_RH, RF_RH |
| 4 | Temperature + calendar-related features + lag features | Prophet_Temp, XGBoost_Temp, SVM_Temp, NN_Temp, RF_Temp |
| 5 | Relative humidity + temperature + calendar-related features + lag features | Prophet_All, XGBoost_All, SVM_All, NN_All, RF_All |

The predictive performance metrics of the 25 multivariate models are summarised and presented in Table 5. For the test period, the forecasts from the best five multivariate models (in terms of predictive accuracy) were compared with the actual values of the $PM_{2.5}$ concentration, as shown in Figure 5. From Table 5, it can be observed that the values of MAE, MASE, and RMSE for the XGBoost_All model are the lowest when compared with those of other multivariate models. The MAE, MASE, and RMSE for the XGBoost_All model are 1.69, 0.77, and 2.3809, respectively. Comparing the univariate and multivariate models, the values of MAE, MASE, and RMSE revealed that the XGBoost_All and RF_All models generated forecasts of $PM_{2.5}$ concentration that were more accurate than those of the ARIMA model (see Tables 2 and 5). These findings suggest that variables (i.e., relative humidity, temperature, time-related features, and lag features) included in the XGBoost_All and RF_All models are good predictors of the concentration of $PM_{2.5}$.

**Table 5.** Predictive performance of the multivariate models.

| Model_id | Model Description | MAE | MASE | RMSE |
|---|---|---|---|---|
| 1 | XGBoost_All | 1.69 | 0.77 | 2.3809 |
| 2 | RF_All | 1.81 | 0.82 | 2.3730 |
| 3 | RF_Temp | 1.82 | 0.83 | 2.3582 |
| 4 | RF_RH | 1.83 | 0.83 | 2.4226 |
| 5 | RF_Lag | 2.02 | 0.92 | 2.4825 |
| 6 | XGBoost_Lag | 2.02 | 0.92 | 2.5920 |
| 7 | XGBoost_Temp | 2.10 | 0.96 | 2.5971 |
| 8 | Prophet_Temp | 2.16 | 0.98 | 2.6942 |
| 9 | RF_Time | 2.26 | 1.03 | 2.7664 |
| 10 | Prophet_Time | 2.30 | 1.04 | 2.9765 |
| 11 | NN_Lag | 2.31 | 1.05 | 2.7369 |
| 12 | Prophet_All | 2.34 | 1.06 | 2.8370 |
| 13 | SVM_Lag | 2.35 | 1.07 | 3.0027 |
| 14 | SVM_Temp | 2.45 | 1.11 | 3.1188 |
| 15 | Prophet_RH | 2.64 | 1.20 | 3.2143 |
| 16 | Prophet_Lag | 2.65 | 1.21 | 3.2254 |
| 17 | SVM_Time | 2.66 | 1.21 | 3.26243 |
| 18 | XGBoost_RH | 2.90 | 1.32 | 3.7129 |
| 19 | SVM_RH | 2.95 | 1.34 | 3.5400 |
| 20 | SVM_All | 2.95 | 1.34 | 3.5315 |
| 21 | NN_RH | 3.35 | 1.52 | 4.6193 |
| 22 | NN_All | 3.73 | 1.70 | 4.3048 |
| 23 | XGBoost_Time | 5.19 | 2.36 | 6.0694 |
| 24 | NN_Temp | 6.07 | 2.76 | 6.6457 |
| 25 | NN_Time | 7.53 | 3.42 | 8.0063 |

*3.4. Ensemble Models*

Ensemble methods utilize a combination of machine learning algorithms to produce better forecasts. Previous research of forecasting problems has shown that ensembles achieve better predictive performance when compared with standalone algorithms [45,50]. In this study, the top three models in terms of predictive performance were incorporated into the ensemble model. The top three models were XGBoost_All, RF_All, and ARIMA (see Tables 2 and 5). The three types of ensemble models were (1) average, (2) median, and (3) weighted (weights were allocated based on predictive performance: XGBoost_All (weight = 3), RF_All (weight = 2), and ARIMA (weight = 1)). The weights were assigned based on the predictive performance of each model.

The predictive performance of the ensemble models, which is quantified through the use of MAE, MASE, and RMSE, is shown in Table 6. A comparison of the forecasts from the ensemble models and the actual concentration values of $PM_{2.5}$ for the test period is presented in Figure 6. It can be seen from the data in Table 6 that the performance metrics (MAE, MASE, and RMSE) of the ensemble (weighted) model are the lowest. Overall, the ensemble models outperform the univariate and multivariate models (see Tables 2, 5 and 6) in terms of predictive performance. The ensemble model developed in this study is a hybrid of the 'best' models (i.e., XGBoost_All, RF_All, and ARIMA), which could be the reason for its improved performance. Interestingly, it was found that the ensemble models developed in this study produced more accurate forecasts of $PM_{2.5}$ concentration.

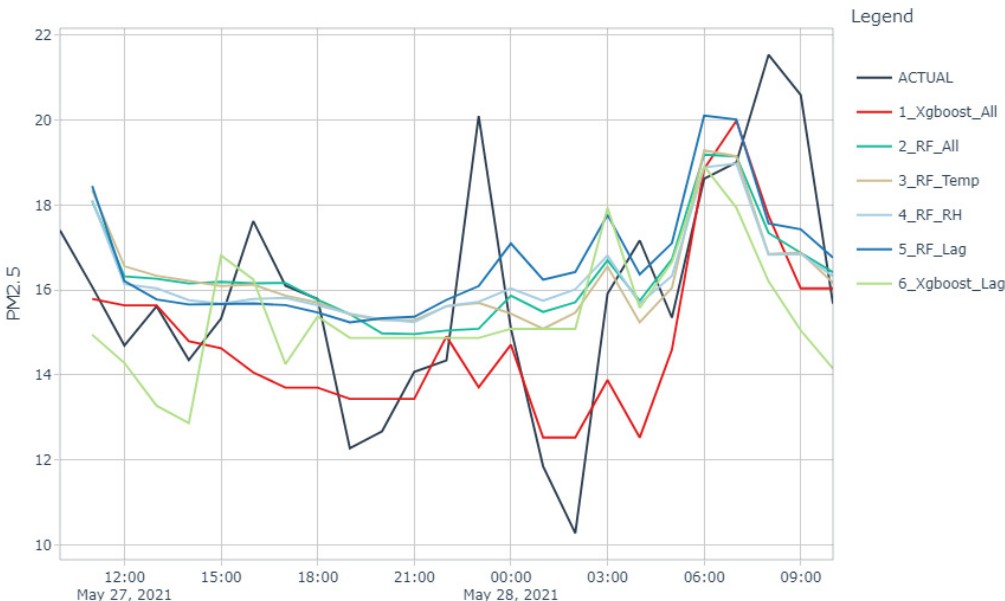

**Figure 5.** Forecasts from the 'best 5' multivariate models compared to the actual values of PM$_{2.5}$ concentration.

**Table 6.** Predictive performance for univariate models.

| S/N | Model | MAE | MASE | RMSE |
|---|---|---|---|---|
| 1 | Ensemble (weighted) | 1.57 | 0.71 | 2.1876 |
| 2 | Ensemble (mean) | 1.60 | 0.73 | 2.1985 |
| 3 | Ensemble (median) | 1.61 | 0.73 | 2.1830 |

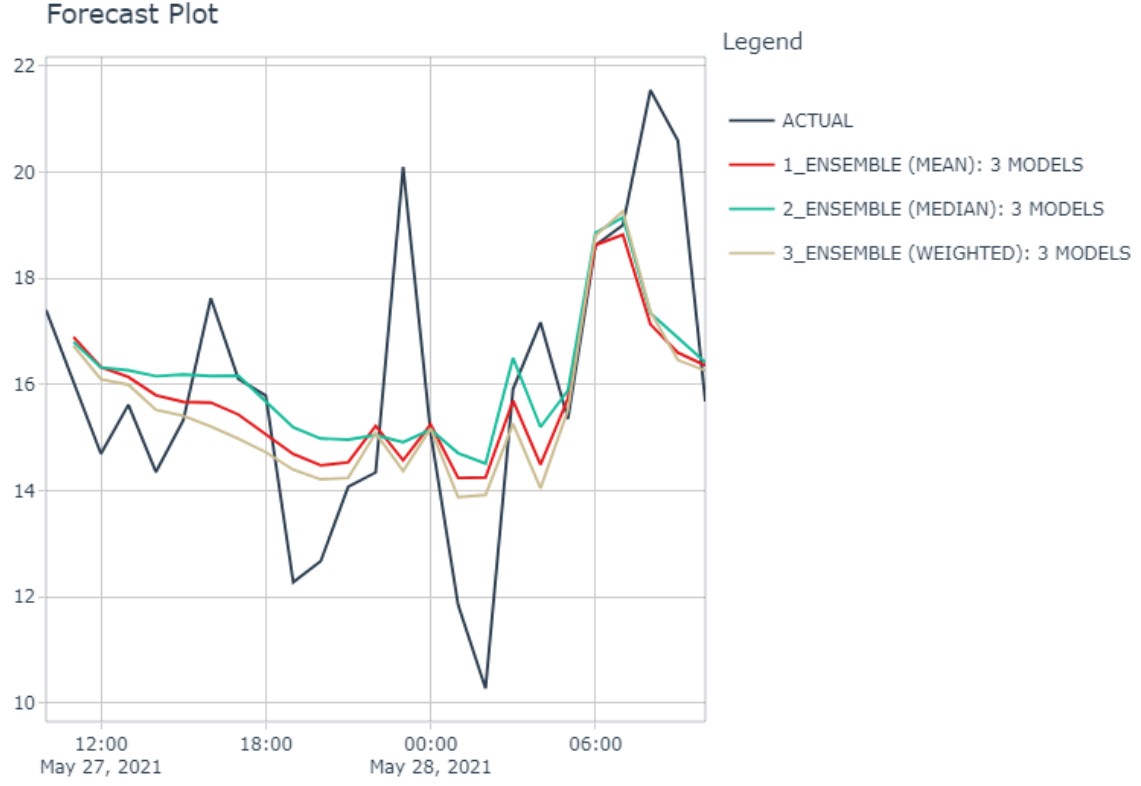

**Figure 6.** Forecast values of PM$_{2.5}$ concentration (ensemble models) compared with the actual values.

## 4. Discussion

The overall goal of this study was to create reliable models for the forecasting of $PM_{2.5}$ concentration levels. The present study showed that the temperature and relative humidity values are good predictors of the $PM_{2.5}$ concentration. The inclusion of these meteorological variables improved the performance of multivariate models, i.e., XGBoost_All and RF_All. One unanticipated finding was that the ARIMA model outperformed some of the multivariate models, e.g., NN_All. In terms of predictive accuracy, the ensemble models (XGBoost_All, RF_All, and ARIMA) outperformed standalone algorithms, such as XGBoost. Taken together, this study showed that the availability of meteorological data is vital for the development of reliable models for forecasting of $PM_{2.5}$ concentrations. Furthermore, the proposed hybrid model, XGBoost-RF-ARIMA, generates a reliable forecast of $PM_{2.5}$ concentrations.

Previous studies have made attempts to identify the 'best' set of predictors for $PM_{2.5}$ concentration levels. The findings detailed in the present study are consistent with those obtained in previous studies. For instance, studies have shown that temperature and relative humidity have influences on the concentration of $PM_{2.5}$ [12,37]. Moreover, a previous study by Wang et al. [51] revealed that variations in the meteorological conditions have significant influences on the concentration of $PM_{2.5}$. Therefore, the availability of data on meteorological factors plays a crucial role in the development of reliable models for forecasting of $PM_{2.5}$ concentrations.

As stated previously, it was somewhat surprising that the ARIMA model outperformed some of the multivariate models. This finding is contrary to those of previous studies, which have shown that nonlinear models (such as neural networks) tend to outperform ARIMA models [50]. This finding could be attributed to several factors: First, this could be due to the unavailability of data on other meteorological variables, such as wind speed; the inclusion of these variables could improve the performance of multivariate models. Second, the availability of data on other factors that affect the concentration of $PM_{2.5}$ could aid in the application of variable selection methods. Research has shown that the application of variable selection methods improves the performance of prediction models [52,53]. The ensemble models outperformed the standalone algorithms. This result is consistent with those reported by Zhou et al. [50], who showed that ensemble models tend to generate better forecasts. These findings suggest that hybrid models constitute useful tools for $PM_{2.5}$ concentration forecasting.

## 5. Conclusions

Reliable forecast models can be useful tools for understanding the factors that can affect the concentration of $PM_{2.5}$. This information can be used to develop strategies and policies for reducing the concentrations of air pollutants such as $PM_{2.5}$. In the present study, seven algorithms and three hybrid models were used for forecasting the hourly concentrations of $PM_{2.5}$. To evaluate the predictive performance of the proposed models, the trained models were used to generate 24-hour-ahead forecasts of $PM_{2.5}$ concentration based on air quality data collected in Lagos, Nigeria. Subsequently, the predictive performance of the models was compared. Two key findings emerged from this study: (1) meteorological factors are useful for the forecasting of $PM_{2.5}$ concentration, and (2) ensemble models (e.g., XGBoost-RF-ARIMA) generate a more reliable forecast of $PM_{2.5}$ concentration when compared with standalone algorithms.

However, the following conclusions can also be drawn: (1) Metrological variables cannot adequately forecast $PM_{2.5}$ concentration at its peak and lowest periods. This finding suggests the need for the collection of data on other factors (such as number of vehicles) affecting the concentration of $PM_{2.5}$. (2) Advancements in data science are providing new tools that can be used for generating reliable forecasts of $PM_{2.5}$ concentration. (3) Finally, meteorological, calendar-based, and time-based variables were used for the development of the multivariate forecast models; however, future studies can collect data on other variables (e.g., number of vehicles, types of vehicles, and other sources of air pollutants).

The additional data can then be used to update the multivariate models. Subsequently, the updated models could be used as a laboratory to test the impacts of policies on air pollutant concentrations. Taken together, this study provides evidence for the effectiveness of using ensemble modelling techniques for air pollution forecasting.

The most important limitation of this study lies in the fact that meteorological factors were the only variables used for model development. Other variables (i.e., calendar- and time-based variables) were derived from the time component. Despite this limitation, the developed models outperformed the naïve model, i.e., the value of MASE was less than 1. This finding indicates that the developed models are reliable. These findings contribute to knowledge in several ways: First, this study shows that metrological and time variables are useful predictors of $PM_{2.5}$ concentration. Second, the study indicates that ensemble modelling techniques can be applied to solve air quality forecasting problems. More research using data on additional variables could provide more evidence on the effectiveness of ensemble models. The proposed models can be used by relevant stakeholders for the forecasting of $PM_{2.5}$ concentration levels.

**Author Contributions:** Conceptualization, O.A.E., O.S.O. and M.O.; data curation, N.E.; formal analysis, O.S.O.; funding acquisition, O.A.E.; investigation, A.S.; methodology, O.A.E., O.S.O. and M.O.; project administration, O.A.E., O.T.B. and O.A.; resources, O.A.; software, O.S.O. and M.O.; visualization, O.S.O.; writing—original draft, O.A.E., O.S.O. and A.S.; writing—review and editing, O.T.B. and N.E. All authors have read and agreed to the published version of the manuscript.

**Funding:** This research effort was funded by The University of Manchester's Global Challenges Research Fund (GCRF) QR grant.

**Institutional Review Board Statement:** Not applicable.

**Informed Consent Statement:** Not applicable.

**Data Availability Statement:** Not applicable.

**Acknowledgments:** The field study was supported by University of Lagos colleagues and students.

**Conflicts of Interest:** The authors declare no conflict of interest.

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
