# Peer review of "Modelling and Forecasting Temporal PM2.5 Concentration Using Ensemble Machine Learning Methods"

_buildings, doi:10.3390/buildings12010046_

Round 1

Reviewer 1 Report

This study compares the PM 2.5 predictive performance of seven algorithms (ARIMA, exponential smoothing, prophet model, neural network, random forest, SVM, XGBoost) based on a built dataset. The data were collected using an air quality sensor, accompanied by meteorological variables (temperature and relative humidity), and were collected over 15-minute intervals over a 6-month period. In addition, other variables were considered such as time-related features, and lag features.

The finding of this study is that the ensemble models such as XGBoost_All, RF_All, and ARIMA outperformed standalone algorithms in terms of predictive accuracy (lines 387-388). In addition, the proposed hybrid model, XGBoost-RF-ARIMA, generates a reliable forecast of PM2.5 (lines 390-391). This finding is necessary to support air pollution forecasts and warnings.

However, there are some issues below that need to be clarified.

  1. Looking at Figures 4 and 5, why is there such a considerable difference in the value of PM2.5 between the actual measurement and the first best method?
  2. Four variables were considered including temperature, relative humidity, time-related features, and lag features (lines 350-351). Is it enough for the actual working model?
  3. How would the predictive performance be improved if two factors such as wind and precipitation are added as suggested by McKendry (2002) in lines 394-396.
  4. An efficient model can also be obtained from the ensemble of the remaining methods with different weights. Meanwhile, the authors only surveyed for three best single models with integer weights (lines 364-367). Please clarify this part.
  5. It is suggested that the authors should compare the accuracy metrics (accuracy, precision, recall, F1) of the machine learning model in addition to the error metrics.

Reviewer 2 Report

The authors have tested the reliability of forecast models for understanding the factors that can affect the concentration of PM2.5.
The paper is well-written and structured.

My suggestions to improve the manuscript are as follows:

  • line 5, check the spacing at the end of the sentence;
  • line 67, this section can be merged in the previous one;
  • line 78, in the method section a workflow diagram should be added showing the methodology steps used to carry out the presented research study;
  • The innovative contribution of the research study should be properly defined in the introduction section and then, emphasize in the conclusion section.
